# Hypoxia-Inducible Factor-1 Alpha Expression Is Predictive of Pathological Complete Response in Patients with Breast Cancer Receiving Neoadjuvant Chemotherapy

**DOI:** 10.3390/cancers14215393

**Published:** 2022-11-02

**Authors:** César L. Ramírez-Tortosa, Rubén Alonso-Calderón, José María Gálvez-Navas, Cristina Pérez-Ramírez, José Luis Quiles, Pedro Sánchez-Rovira, Alberto Jiménez-Morales, MCarmen Ramírez-Tortosa

**Affiliations:** 1Pathological Anatomy Service, University Hospital San Cecilio, Parque Tecnológico de la Salud (PTS), Avda. del Conocimiento, 18016 Granada, Spain; 2Medical Oncology Service, Complejo Hospitalario de Jaén, Avda. del Ejército Español 10, 23007 Jaén, Spain; 3Pharmacogenetics Unit, Pharmacy Service, University Hospital Virgen de las Nieves, Avda. de las Fuerzas Armadas 2, 18004 Granada, Spain; 4Department of Biochemistry and Molecular Biology II, Faculty of Pharmacy, Campus Universitario de Cartuja, Universidad de Granada, 18011 Granada, Spain; 5Department of Physiology, Faculty of Pharmacy, Campus Universitario de Cartuja, University of Granada, 18011 Granada, Spain

**Keywords:** hypoxia-inducible factor 1, breast cancer, neoadjuvant chemotherapy, prognostic factor, pathological complete response

## Abstract

**Simple Summary:**

Standard neoadjuvant chemotherapy, based on taxanes and anthracyclines, makes conservative treatment of breast cancer possible and it allows for the evaluation of the tumor response in terms of achieving pathological complete response. Whereas hypoxia participates in carcinogenesis, resulting in less differenced tumor cells and poorer prognosis, HIF-1α could be predictive of the tumor response to treatment. Nonetheless, very few studies have evaluated the predictive value of HIF-1α in breast cancer in patients receiving neoadjuvant chemotherapy.

**Abstract:**

To demonstrate the value of hypoxia-inducible factor-1α (HIF-1α) in predicting response in patients with breast cancer receiving standard neoadjuvant chemotherapy (NAC). Methods: Ninety-five women enrolled in two prospective studies underwent biopsies for the histopathological diagnosis of breast carcinoma before receiving NAC, based on anthracyclines and taxanes. For expression of HIF-1α, EGFR, pAKT and pMAPK, tumor samples were analyzed by immunohistochemistry in tissues microarrays. Standard statistical methods (Pearson chi-square test, Fisher exact test, Kruskal–Wallis test, Mann–Whitney test and Kaplan–Meier method) were used to study the association of HIF-1α with tumor response, survival and other clinicopathologic variables/biomarkers. Results: HIF-1α expression was positive in 35 (39.7%) cases and was significantly associated to complete pathological response (pCR) (*p* = 0.014). HIF-1α expression was correlated positively with tumor grade (*p* = 0.015) and Ki-67 expression (*p* = 0.001) and negativity with progesterone receptors (PR) (*p* = 0.04) and luminal A phenotype expression (*p* = 0.005). No correlation was found between HIF-1α expression and EGFR, pAKT and pMAPK. In terms of survival, HIF-1α expression was associated with a significantly shorter disease-free survival (*p* = 0.013), being identified as an independent prognostic factor in multivariate analysis. Conclusions: Overexpression of HIF-1α is a predictor of pCR and shorter DFS; it would be valuable to confirm these results in prospective studies.

## 1. Introduction

Standard neoadjuvant chemotherapy (NAC), based on a schedule of anthracyclines and taxanes, is the treatment of choice for locally advanced breast tumors and inflammatory carcinomas [1]. The administration of NAC not only makes conservative treatment possible, but also precision medicine according to its efficacy [1] and the evaluation of the pathologic response of the tumor in terms of achieving pathologic complete response (pCR) with rates ranging from 3% to 48% and a partial response with a rate of 61.2% [2,3]. It has been shown that pCR is a prognostic factor for disease-free survival (DFS) and overall survival (OS) [4,5], probably because it reflects the eradication of micrometastatic disease [5]. For this reason, predictive markers of response identification have been a topic of study for a long time; estrogen (ER) and progesterone receptors (PR) and human epidermal growth receptor 2 (HER2) status are the best known of these markers.

Multiple factors affect cancer development [6,7]. When a situation of hypoxia develops in the tumor microenvironment during the process of neoplastic progression, cells with more aggressive tumor phenotypes, higher mutation rates and increased metastatic potential are selected [8]. The HIF-1α (hypoxia-inducible factor 1α) transcription factor seems to be the key molecular complex in the cellular response to hypoxia [8,9]. Furthermore, the synthesis of HIF-1α can be regulated for other mechanisms, independent of tissue oxygenation, across activation of the phosphatidylinositol 3-kinase (PI3K/AKT) and mitogen-activated protein kinase (Ras/Raf/MAPK) pathways [10,11]. These pathways can be activated by receptors with tyrosine kinase activity, such as the epidermal growth factor receptor (EGFR) [12].

Several studies have examined the role of HIF-1α as a prognosis factor in breast cancer and have associated HIF-1α overexpression with shorter DFS and OS [13]. However, little is known about the predictive value of HIF-1α response in breast cancer. To date, few published papers have demonstrated the relation between HIF-1α overexpression and pCR after treatment with NAC based on anthracyclines and taxanes. The objectives of our study were to demonstrate the value of HIF-1α in predicting response in patients diagnosed with breast cancer and given an NAC schedule of anthracyclines and taxanes, to study the relation between HIF-1α overexpression and other clinicopathologic variables of well-established predictive value and, finally, to study the intracellular signaling pathways involved in HIF-1α regulation and to analyze the potential prognostic value of HIF-1α.

## 2. Materials and Methods

### 2.1. Patients and Treatment Management

The study included 95 patients diagnosed with stage II-III breast cancer who received neoadjuvant chemotherapy at Complejo Hospitalario de Jaén. All patients were participants in two prospective phase 2 studies. In study A, 73 patients received 3 cycles of epirubicin (90 mg/m^2^) and cyclophosphamide (600 mg/m^2^), followed by 6 cycles of paclitaxel (150 mg/m^2^) and gemcitabine (2500 mg/m^2^), with or without trastuzumab (2 mg/kg/week, with a loading dose of 4 mg/kg) in accordance with HER2 status [14]. In study B, 22 patients received 4 cycles of doxorubicin (60 mg/m^2^) and cyclophosphamide (600 mg/m^2^) followed by 4 cycles of docetaxel (100 mg/m^2^) [15]. Previous axillar status to chemotherapy was firstly evaluated using sonography. Suspicious cases were confirmed by needle core biopsy. All women underwent surgery after cytostatic treatment. Modified radical mastectomy or conservative surgery was performed according to surgeons’ criteria. Patients with cN0 were submitted to axillary intraoperative study through sentinel lymph node biopsy. Axillary lymphadenectomy was performed in cN+ patients. Patients who underwent conservative surgery also received radiotherapy. All patients with hormone-receptor-positive tumors were treated with hormonal therapy for 5 years. The median follow-up of patients was 7.4 years. The patients’ characteristics are shown in Table 1.

### 2.2. Histology and Response Pathological Evaluation

Histological examinations were performed on slides stained by hematoxylin–eosin from those that were paraffin embedded. Histological grade was determined according to the modified Bloom–Richardson classification [16]. pCR was defined as the absence of invasive carcinoma in the breast and lymph nodes according to the Miller–Payne criteria. Additionally, the single presence of carcinoma in situ was equally considered as pCR [17].

### 2.3. Tissue Microarray Construction

Hematoxylin-and-eosin-stained sections from core biopsies (pretreatment) and surgical specimens (post-treatment) were marked on individual paraffin blocks. Two tissue cores (1.5 mm in diameter) were obtained from each specimen. Additionally, other tissues, both non-neoplastic and neoplastic samples, were included as controls following the Kononen methodology [18]. A hematoxylin-and-eosin-stained section was reviewed to confirm the presence of morphologically representative areas of the original lesions.

### 2.4. Immunohistochemistry

The immunohistochemical analysis was blinded. The sections of tissue were deparaffinized with xylene and hydrated in gradient alcohols. After the deparaffinization of tissue sections, antigen retrieval was performed with the PTLink module (Dako, Glostrup, Denmark) using Dako pH Antigen Retrieval fluid (Dako) followed by several washes in water. They were then placed onto an Autostainer Plus Link (Dako, Demark) where the remainder of the immunohistochemical staining was performed using Envision FLEX (DAKO). Briefly, sections were first placed in washing buffer followed by blockade of endogenous peroxidase with 3% hydrogen peroxide for 5 min. Then, the primary antibody ER (rabbit monoclonal antibody, prediluted, clone SP1 Master Diagnostica), PR (rabbit monoclonal antibody, prediluted, clone Y85 Master Diagnostica), Ki-67 (rabbit monoclonal antibody, prediluted, clone SP6 Master Diagnostica), HIF-1α (mouse monoclonal antibody, diluted 1:50, Becton-Dickinson Biosciences, Palo Alto, CA, USA), pAKT (rabbit monoclonal antibody, diluted 1:25, clone 736E11 Cell Signaling Technology, Beverly, MA, USA) and pMAPK (rabbit monoclonal antibody, diluted 1:100 clone 20G11 Cell Signaling echnology) were applied for 30 (ER), 20 (PR), 30 (Ki-67), 120 (pAKT) and 60 (pMAPK) minutes at room temperature, except HIF-1α, which was applied overnight at 4 °C. Sections were then treated with immunodetection solution consisting of biotinylated secondary antibody for 30 min. Diluted 1:50 liquid 3,3’-diaminobenzidine (Dako) was used as a chromogenic agent and sections were counterstained in Meyer’s hematoxylin. As a negative control, the primary antibody was replaced by a non-immune serum.

HER2 status was determined using the Dako HERceptest (Dako Denmark A/S, Glostrup, Denmark) as well as a fluorescence in situ hybridization test in biopsy specimens with a 2+ score via IHC analysis. EGFR expression was determined using the Dako EGFR pharma (Dako Denmark A/S, Glostrup, Denmark).

### 2.5. Evaluation of Immunohistochemical Staining

For ER and PR, two approaches were used. All red method scoring was used for assessing staining intensity and the percentage of positive cells. The total score is obtained by adding the staining score and intensity score. Any score between 0 and 2 is considered ER or PR negative; any score above 2 is considered ER or PR positive [19]. A case was considered positive when staining for ER and PR was found in 10% or more of tumor cells [20]. Tumors were considered to have high rates of proliferation according to the Ki-67 labeling index if 20% of cell nuclei stained positive for Ki-67 [21]. HER2-positive cases were defined as having membrane staining score of +3 or +2 with gene amplification by FISH [22].

HIF-1α was scored only according to the presence (1+) or absence (0) of nuclear expression: at least 5% of cells had to be stained to be considered positive [13,23]. However, pAKT and pMAPK were scored according to the presence (1+) or absence (0) of nuclear and/or cytoplasmic expression: the cutoff value was 10% [24]. For EGFR, all cells that exhibited some membrane staining were considered positive.

### 2.6. Statistical Analysis

Statistical analysis was carried out using SPSS version 27.0 software (SPSS Inc., Chicago, IL, USA) (SPSS IBM Statistics 27.0 for Windows). The Pearson chi-square test/Fisher exact test was used to study the association between pCR and HIF-1α with clinicopathologic variables. The association between protein expression and pCR was studied using nonparametric tests (Kruskal–Wallis/Mann–Whitney) and the Pearson chi-square test/Fisher exact test. Multivariate logistic regression was used to examine the predictors of pCR. The relation between the expressions of different proteins was studied using the Fisher exact test. Finally, survival was analyzed using the Kaplan–Meier method, with determination of significance using the long rank test. Multivariate analysis was carried out using Cox regression analysis. Data analysis is reported according to REMARK guidelines [25].

Probability (*p*) values of less than 0.05 were considered statistically significant.

## 3. Results

### 3.1. Relation between HIF-1α Expression and pCR

Out of 95 samples analyzed, HIF-1α expression was determined in 88 (92.6%). Of these, 35 (39.72%) were considered positive (Figure 1). The relation between HIF-1α and pCR was examined by studying the HIF-1α variable, both quantitatively as a percentage (%) and qualitatively as a dichotomy (≥5%). A statistically significant relation was found between HIF-1α expression and pCR: patients whose tumors overexpressed HIF-1α were more likely to achieve pCR (Table 2).

### 3.2. Relation between HIF-1α Expression and Biological Markers

A positive relationship between HIF-1α expression and Grade (*p* = 0.015) and Ki-67 (*p* = 0.001) was identified. HIF-1α expression was negatively correlated with PR (*p* = 0.049) and Luminal A phenotype (*p* = 0.005) (Table 3).

### 3.3. Relation between HIF-1α Expression and pATK, pMAK and EGFR

In 95 samples analyzed, pAKT expression was determined in 81 (85.3%) patients and it was considered positive in 57 (70.37%) (Figure 1). Using pMAPK, expression was determined in 74 (77.9%) patients and it was considered positive in 61 (82.43%) (Figure 1). For EGFR, expression was determined in 88 (92.6%) patients: of these, none were considered positive because there was no membrane staining of any tumor cells (Figure 1). No relation was found between the expression of these proteins and HIF-1α (Appendix A).

### 3.4. Predictive Factors of Response to Treatment—Multivariate Analysis

For the resulting model consisting of the variables Ki-67, HIF-1α and molecular phenotype, only basal phenotype was an independent predictive factor of pCR (*p* = 0.001) (Appendix A).

### 3.5. Sulvival Analysis—Prognostic Markers

In univariate analysis, the markers associated with shorter DFS were: HIF-1α positive (*p* = 0.013) (Figure 2), Ki-67 positive (*p* = 0.002), basal phenotype (*p* = 0.001), pAKT negative (*p* = 0.009) and ER negative (*p* = 0.024, *p* = 0.010). As for OS, markers associated with decreased survival were HIF-1α positive (a trend that did not reach statistical significance, *p* = 0.08), Ki-67 positive (*p* = 0.022), basal phenotype (*p* = 0.007), pAKT negative (*p* = 0.007) and ER negative (*p* = 0.001).

In multivariate analysis with correlation of ER, grade, Ki-67, pCR, HIF-1α and pAKT, it was shown that pCR and the expression of Ki-67, HIF-1α and pAKT were independent predictors of DFS, while ER and pAKT were independent prognostic factors of OS (Table 4).

## 4. Discussion

To date, just a few studies have demonstrated the existence of statistically significant relations between the expression of HIF-1α and pCR in breast cancer after neoadjuvant chemotherapy based on anthracyclines and taxanes [26,27,28]. In our study, more than a third of patients were considered HIF-1α positive. These results concur with other findings reported in the literature from studies using the same cutoff point as in our study [13,23,29]. However, the rate of positivity in the studies that examined HIF-1α in breast cancer ranges from 1% to 80.2%, probably due to the use of different cutoff points and assessment systems [11].

Among the studies where the predictive value of HIF-1α in breast cancer is evaluated, very few have investigated the relation between the expression of HIF-1α and pCR or the predictive value of the molecule in breast cancer patients going under neoadjuvant chemotherapy based on anthracyclines and taxanes [26,27,28,29,30,31]. In the study published by Yamamoto et al. (2008), of all of the patients who achieved pCR, 100% were positive for HIF-1α, whereas only 66.7% of the other patients were. The difference did not reach statistical significance, probably because of the small study sample [29]. Another two studies conducted by Generali et al. analyzed the predictive value of HIF-1α. The first study is a phase 2 clinical trial where patients were randomized to receive neoadjuvant anthracycline-based chemotherapy versus anthracycline plus tamoxifen [30]. Overexpression of HIF-1α was associated with lower clinical response. However, no statistically significant relation was found between pCR and overexpression of HIF-1α, which makes the above findings questionable. Moreover, staining intensity was used to establish the cutoff point for HIF-1α (negative: 0, +1 vs. positive: +2). Of the five patients who achieved pCR, only one had no staining; the other four were classified as +1. In many other studies, including ours, the cutoff was the percentage of cells stained; therefore, weak staining, considered negative for HIF-1α in this study, could be considered positive by other authors, depending on the percentage of stained cells. The other study that evaluated the predictive value of HIF-1α was also a phase 2 trial, where patients were randomized to receive either letrozole or letrozole plus oral cyclophosphamide [31]. Overexpression of HIF-1α was associated with lower rates of clinical response, with no reference to the relation with pCR rate. In a recent meta-analysis, it was shown that pCR correlates with improved DFS and OS, pointing to the lack of prognostic value of the clinical responses [5]. In our study, pCR was an independent prognostic factor of longer DFS and almost significant for OS (*p* = 0.06).

Recent studies continue this evaluation. According to the study published by Tiezzi et al. (2013), an overall reduction in HIF-1α and HIF-2α expression was observed in patients after using NAC based on anthracycline and taxanes, but no association was observed between HIF-2α expression and its predictive value of pCR. However, the pCR rate in HIF-1α-negative patients was 5%, whereas in HIF-1α-positive patients, it was 21% (*p* = 0.03) [28]. These results are consistent with those in our study. Furthermore, in a cohort formed of 220 patients who received a treatment regime based on anthracyclines and taxanes, 68.2% were considered HIF-1α positive (150/220). Otherwise, in this case, HIF-1α-negative patients had a higher pCR rate rather than HIF-1α-positive patients (*p* = 0.027) [27]. In the most recent study, the expression of HIF-1α was identified in 104 tumor biopsies. Thus, more than a third of the patients were considered HIF-1α positive. The evaluation of the predictive value of HIF-1α showed significant association with resistance and favorable response to NAC based on anthracyclines and taxanes (*p* < 0.001). Specifically, patients with a lower expression of HIF-1α were in the favorable-response group while those in the resistance group had a higher expression of the molecule [26].

Other markers have more predictive value. In our study, we found that patients with negative hormone receptors, more undifferentiated tumors, a higher rate of proliferation and basal phenotype were significantly related to higher pCR rate. These results concur with those described in the literature [3,32,33].

An attempt was made to explain the paradoxical observation that some of the variables were predictive of both unfavorable prognosis and chemosensitivity. Some authors have related these discrepant findings with attaining pCR or not. That is to say that it seems clear that patients with a triple-negative phenotype who receive NAC and do not reach pCR, despite good clinical response, have a less prolonged survival. However, those who attain pCR apparently have an excellent prognosis [34,35].

On the other hand, we found a statistically significant relation between HIF-1α expression and hormone receptor negativity. It has been demonstrated that hypoxia decreases ER and PR levels in breast cancer, suggesting a relation between HIF-1α expression and resistance to hormonal therapy [36]. We also found a statistically significant positive relation between HIF-1α expression and the proliferation marker Ki-67. These results are consistent with the findings reported in the literature, as it has been suggested that the higher proliferation rate of tumor cells causes HIF-1α activation [37]. The expression of HIF1-α has also been linked with poorly differentiated tumors [37,38]. Our findings also support these statements; it is considered that hypoxia induces genetic alterations, promoting morphological changes in the cell itself and in its nucleus, resulting in more undifferentiated tumor cells [7,39].

The third objective of our study was to determine whether the activation of HIF-1α is independent of tissue oxygen concentration, so we studied the molecular pathways that might be involved. One of the membrane receptors related with HIF-1α activation in normoxic conditions is EGFR [12]. Jögi et al. found a significant relationship between the expression of EGFR and HIF-1α [40]. We did not find EGFR expression in any of the patients in the study, although the external controls were positive, suggesting that there may be other receptor tyrosine kinases capable of inducing the transcriptional activity or increasing HIF-1α stability under normoxic conditions but ROS dependent [41]. Some authors have elucidated a role for mitochondrial-generated ROS in tumoral HIF-1α stabilization [9,42]. On the basis of the foregoing, this pathway could be the reason why HIF-1α expression incremented in this study due to the increase in oxidative stress status in breast cancer patients who received neoadjuvant chemotherapy treatment, as described in previous investigations of our group [43].

Once the membrane receptor is stimulated, there are several intracellular signaling pathways that are associated with the synthesis of HIF-1α under normoxic conditions. These pathways include the PI3K/AKT/mTOR and RAS/RAF/MAPK pathways. We found overexpressed pAKT in 70% of cases, which is consistent with the results of other published studies [24,44]. The relation between the expression of HIF-1α and pAKT was examined by immunohistochemistry in breast cancer in a single study. Gort et al. [44] studied the overexpression of both proteins in 95 patients and concluded that low pAKT expression correlated significantly with low HIF-1α expression. We found no relation between the two molecules in our study. With respect to the RAS/RAF/MAPK pathway, we detected positivity in 83%, as reported in other published studies [45,46]. No significant association was found between overexpression of HIF-1α and pMAPK. Kronblad et al. [45] studied the expression of pMAPK and HIF-1α in 21 samples of ductal carcinoma in situ breast cancer, finding a positive relation between the expression of both molecules; the authors emphasized the overexpression of pMAPK in less-hypoxic areas. These results are consistent with the ones shown in the study conducted by Hsu et al. (2016) [46]. These findings and our results support activation of HIF-1α mediated by the hypoxic conditions existing in the tumor microenvironment.

As for the prognostic value of HIF-1α, in our study, we saw how overexpression of this protein was an independent prognostic factor for shorter DFS. Our results confirm those of other authors included in a meta-analysis of 5177 patients who used the same cutoff value as in our study [37].

Interestingly, in our study, we also found that pAKT overexpression was an independent prognostic factor of more prolonged DFS and OS. The largest study to analyze the prognostic value of pAKT was carried out by Yang et al. [47] with 1202 patients enrolled in the study of neoadjuvant NSABP B-28; patients who overexpressed pAKT had a longer DFS compared to patients who did not overexpress this protein.

## 5. Conclusions

It was shown that overexpression of HIF-1α is a predictor of pCR and shorter DFS in patients who received a neoadjuvant chemotherapy schedule based on anthracyclines and taxanes. In addition, HIF-1α is related to other variables with a more consolidated predictive and prognostic value. All these variables are associated with a more aggressive and hypoxic tumor microenvironment. We believe that it would be interesting to confirm these results in prospective studies, given the need for expanding the small panel of predictive markers in breast cancer.

## Figures and Tables

**Figure 1 cancers-14-05393-f001:**
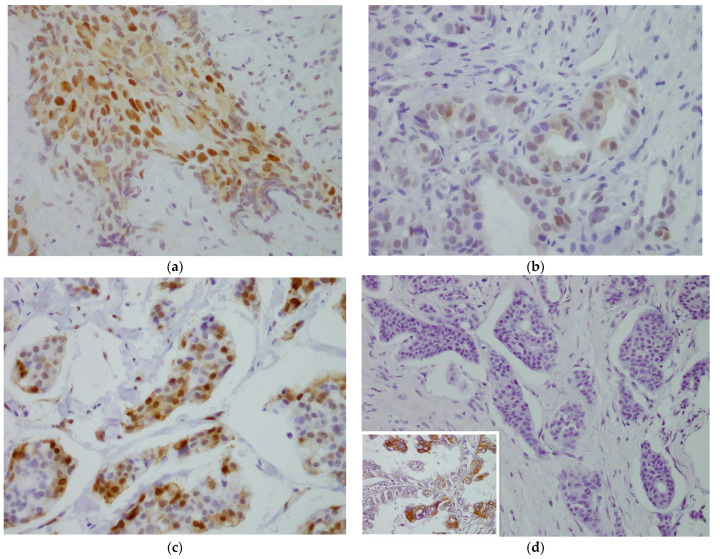
Evaluation of immunohistochemical staining. (**a**) For HIF-1α, moderate staining of nuclei and slight staining of some cytoplasmic areas, >5% in tumor cells. 40×. (**b**) For pAKT, mild-to-moderate nuclear and cytoplasmic staining, ≥10% in tumor cells. 40×. (**c**) For pMAPK, strong nuclear staining and mild-to-moderate cytoplasmic staining, >10% in tumor cells. 40×. (**d**) For EGFR, negative membrane staining. A staining positive control for EGFR of lung cancer was inserted in the image 40×.

**Figure 2 cancers-14-05393-f002:**
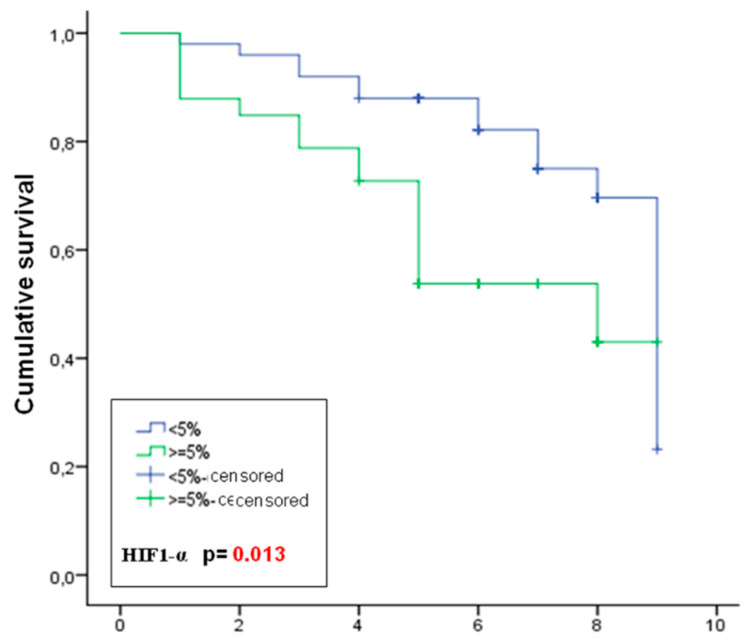
Kaplan–Meier curves of disease-free survival of patients stratified by HIF-1α expression.

**Table 1 cancers-14-05393-t001:** Clinicopathologic characteristics of patients.

Characteristics	Number of Cases (%)
Age at diagnosis (years)	
<40	19 (20%)
40–49	38 (40%)
50–59	18 (18.9%)
>60	20 (21.1%)
Mean	20 (21.1%)
Range	27–74
Pretreatment tumor size (cm)	
1–1.9	4 (4.2%)
2–2.9	22 (23.2%)
3–3.9	22 (23.2%)
4–4.9	17 (17.9%)
>4.9	23 (24.2%)
Not measurable	7 (7.4%)
Histological type	
Ductal infiltrating	72 (75.8%)
Lobular infiltrating	9 (9.5%)
Inflammatory	6 (6.3%)
Mucinous	6 (6.3%)
Mixed	2 (2.1%)
Histological grade	
1	20 (21%)
2	37 (39%)
3	34 (35.8%)
Not evaluable	4 (4.2%)
Clinical TNM at diagnosis	
T	
T1	6 (6.3%)
T2	62 (65.3%)
T3	16 (16.8%)
T4	8 (8.4%)
Tx	3 (3.2%)
N	
N0	34 (35.8%)
N1	46 (48.4%)
N2	15 (15.8%)
N3	0 (0%)
Nx	0 (0%)
M	
M0	95 (100%)
M1	0 (0%)
Clinical stage	
IIA	32 (33.7%)
IIB	31 (32.6%)
IIIA	21 (21.1%)
IIIB	8 (8.4%)
Not evaluable	3 (3.2%)
ER	
≥10%	69 (72.6%)
<10%	25 (26.3%)
Not evaluable	1 (1.1%)
Count ≥ 3	71 (74.7%)
Count < 3	23 (24.2%)
Not evaluable	1 (1.1%)
PR	
≥10%	54 (56.8%)
<10%	40 (42.1%)
Not evaluable	1 (1.1%)
Count ≥ 3	58 (61.1%)
Count < 3	36 (37.1%)
Not evaluable	1 (1.1%)
HER2	
Positive	20 (21.1%)
Negative	72 (75.8%)
Not evaluable	3 (3.1%)
Ki-67	
≥20%	38 (40%)
<20%	56 (58.9%)
Not evaluable	1 (1.1%)
Phenotype	
Basal	13 (13.7%)
HER2	20 (21%)
Luminal A	31 (32.6%)
Luminal B	28 (29.5%)
Not evaluable	3 (3.2%)
Type of surgery	
Conservative	38 (40%)
Not conservative	56 (58.9%)
Not evaluable	1 (1.1%)
Pathological response (M&P)	
Grade 1	8 (8.4%)
Grade 2	22 (23.1%)
Grade 3	28 (29.5%)
Grade 4	17 (17.9%)
Grade 5 (pCR)	20 (21.1%)

Abbreviations: ER, estrogen receptor; PR, progesterone receptor.

**Table 2 cancers-14-05393-t002:** Relation between HIF-1α expression and pCR.

	Number of Cases (%)		* p *
	pCR	No pCR	
HIF-1α < 5%	6 (33.3%)	47 (67.1%)	* 0.014 * ^a^
HIF-1α ≥ 5%	12 (66.7%)	23 (32.9%)	
HIF-1α%	18 (20.5%)	70 (79.5%)	* 0.017 * ^b^
	x = 10.42; SD = 9.53	x = 5.05; SD = 7.52	

^a^ Fisher exact test. ^b^ Mann–Whitney U test. Abbreviations: pCR, pathological complete response; x, mean; SD, standard deviation. Italic means the relation is significant.

**Table 3 cancers-14-05393-t003:** Relation between HIF-1α expression and clinicopathological variables.

Variable		HIF-1α < 5%	HIF-1α ≥ 5%	* p *
Grade 1		15 (83.3%)	3 (16.7%)	* 0.015 * ^ a ^
Grade 2		22 (62.8%)	13 (37.2%)	
Grade 3		13 (41.9%)	18 (58.1%)	
Ki-67 < 20%		29 (80.6%)	7 (19.4%)	* 0.001 * ^b^
Ki-67 ≥ 20%		23 (45.1%)	28(54.9%)	
HER2 −		43 (63.2%)	25 (36.8%)	0.593 ^b^
HER2 +		10 (55.6%)	8 (44.4%)	
ER < 10%		10 (43.5%)	13 (56.5%)	0.080 ^b^
ER ≥ 10%		43 (67.2%)	21 (32.8%)	
ER count < 3		9 (42.9%)	12 (57.1%)	0.072 ^b^
ER count ≥ 3		44 (66.7%)	22 (33.3%)	
PR < 10%		18 (48.6%)	19 (51.4%)	* 0.049 * ^b^
PR ≥ 10%		35 (70%)	15 (30%)	
PR count < 3		16 (47.1%)	18 (52.9%)	* 0.044 * ^b^
PR count ≥ 3		37 (69.8%)	16 (30.2%)	
Phenotype	Basal	5 (41.7%)	7 (58.3%)	
	HER2	10 (55.6%)	8 (44.4%)	
	Luminal A	26 (86.7%)	4 (13.3%)	* 0.005 * ^b^
	Luminal B	12 (46.2%)	14 (53.8%)	

^a^ Pearson chi-square. ^b^ Fisher exact test. Abbreviations: ER, estrogen receptor; PR, progesterone receptor. Italic means the relation is significant.

**Table 4 cancers-14-05393-t004:** Multivariate analysis (Cox regression) of prognostic factors for DFS and OS.

		DFS			OS	
	HR	95% CI	*p*-Value	HR	95% CI	*p*-Value
Grade 1 vs. 2 vs. 3	-	-	0.560	-	-	0.977
Ki-67 ≥ 20% vs. <20%	0.8	1.8–35.5	*0.006*			0.260
ER ≥ 10% vs. <10%	-	-	0.072	3.2	1.3–7.7	*0.008*
pCR vs. no pCR	0.2	0.0–0.6	*0.009*	-	-	0.06
HIF-1α ≥ 5% vs. <5%	2.5	1.0–6.2	*0.047*	-	-	0.295
pAKT ≥ 10% vs. <10%	2.4	1.0–5.9	*0.039*	2.4	1.0–5.7	*0.046*

Abbreviations: DFS, disease-free survival; ER, estrogen receptor; HR, Hazard ratio; OS, overall survival; Italic means the *p*-value is significant.

## Data Availability

Not applicable.

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
