# Peer review of "Hypoxia-Inducible Factor-1 Alpha Expression Is Predictive of Pathological Complete Response in Patients with Breast Cancer Receiving Neoadjuvant Chemotherapy"

_cancers, 2022, doi:10.3390/cancers14215393_

Round 1

Reviewer 1 Report

Ramírez Tortosa report that HIF-1α expression was significantly associated to complete pathological response (pCR) and shorter DFS to NAC based on anthracyclines and taxanes in patients with stage II-III breast cancer that underwent surgery after the cytostatic treatment. The results are interesting showing  that pCR does not predict DFS. This is very important because some trials use pCR as endpoint for accelerating drug approval. However, the criteria used to define pCR was not included in the paper. In particular it is important to know whether in situ carcinomas were included as well as how axillary lymph nodes were evaluated (sentinel lymph node vs sonography).

Reviewer 2 Report

The manuscript is well-written and documented. I have some questions regarding HIF1 expression and EGFR expression.

1. There is much evidence that suggests a positive correlation between HIF1a and EGFR expression.

2. if HIF 1a is not regulated through EGFR then which molecular pathway playing role in the activation of HIF1a.

3. TCGA database shows that HIF1a expression is downregulated but your study is downregulated. please explain.
